# Vasopressin V1a receptor and oxytocin receptor regulate murine sperm motility differently

Hiroyoshi Tsuchiya[1] , Masakatsu Fujinoki[2], Morio Azuma[1], Taka-aki Koshimizu[1]

**Specific receptors for the neurohypophyseal hormones, arginine vasopressin (AVP) and oxytocin, are present in the male reproductive organs. However, their exact roles remain unknown. To elucidate the physiological functions of pituitary hormones in male reproduction, this study first focused on the distribution and function of one of the AVP receptors, V1a. In situ hybridization analysis revealed high expression of the *Avpr1a* in Leydig cells of the testes and narrow/clear cells in the epididymis, with the expression pattern differing from that of the oxytocin receptor (OTR). Notably, persistent motility and highly proportional hyperactivation were observed in spermatozoa from V1a receptor–deficient mice. In contrast, OTR blocking by antagonist atosiban decreased hyperactivation rate. Furthermore, AVP stimulation could alter the extracellular pH mediated by the V1a receptor. The results highlight the crucial role of neurohypophyseal hormones in male reproductive physiology, with potential contradicting roles of V1a and OTR in sperm maturation. Our findings suggest that V1a receptor antagonists are potential therapeutic drugs for male infertility.**

## Introduction

Arginine vasopressin (AVP), a neurohypophyseal hormone, is closely related to oxytocin (OT) (Gimpl & Fahrenholz, 2001; Pierzynski, 2011; Koshimizu et al, 2012; Dumais & Veenema, 2016). Both AVP and OT receptors belong to a family of G protein–coupled receptors, and there are three AVP receptor subtypes: V1a, V1b, and V2 (Koshimizu et al, 2012). Notably, AVP and OT differ structurally by only two amino acids, and their receptors, excluding V2, couple with the same G protein type, Gq, and have 42–55% homology in humans. Recent studies have also demonstrated crosstalk between the OT/OTR and the AVP/V1a signaling pathways (Chini & Manning, 2007; Song & Albers, 2018; Stadler et al, 2020). Multiple studies have highlighted the roles of neurohypophyseal hormones in female reproductive physiology, including our previous study (Clerget et al,

1997; Gimpl & Fahrenholz, 2001; Arrowsmith & Wray, 2014; Tsuchiya et al, 2020). Similarly, the effects of these hormones on male reproductive organs have been investigated (Gimpl & Fahrenholz, 2001; Stadler et al, 2020), and their significant expression was observed (Maggi et al, 1987; Knickerbocker et al, 1988; Fillion et al, 1993; Einspanier & Ivell, 1997; Assinder et al, 2000). However, in contrast to the plethora of information on the OTR distribution in the organs (Maggi et al, 1987; Einspanier & Ivell, 1997; Frayne & Nicholson, 1998; Whittington et al, 2001; Filippi et al, 2002), the cells in male reproductive tissues that express the AVP receptor remain largely unknown (Maggi et al, 1987; Phillips et al, 1990; Mewe et al, 2007; Gupta et al, 2008). Furthermore, only a few studies have analyzed the relationship between AVP and male reproductive physiology (Pierzynski, 2011; Kwon et al, 2013). Corelation studies are challenging because of the high homology between the receptors and the poor selectivity of the commercially available antibodies. In the present study, we were able to overcome such obstacles using a deficient mouse model and an improved in situ hybridization analysis system.

Since the first phenotype of impairment of spatial memory in V1a receptor–deficient mice was reported, various abnormalities have been identified, including cardiovascular system abnormalities, such as decreased heart rate and hypotension, impaired social interaction, impaired glucose homeostasis, insufficient water metabolism, and circadian rhythm misalignment (Egashira et al, 2004, 2007; Koshimizu et al, 2006; Aoyagi et al, 2007, 2009; Yamaguchi et al, 2013). In reproductive physiology, our previous studies have shown that V1a receptor–deficient mice exhibit reduced uterine contractions, delayed parturition, and decreased litter size (Tsuchiya et al, 2020).

Spermatozoa are produced in the testes, followed by their migration downstream to the epididymis (Knobil & Neill, 2006; Borg et al, 2010), which is a long convoluted tubule divided into nine compartments in mice (Sullivan & Mieusset, 2016). Empirically, it has been roughly divided into four regions: the initial segment (IS), caput, corpus, and cauda. While the proximal parts, the IS and caput, are crucial for the sperm to acquire motility and fertilization capacity, the cauda epididymis is crucial for sperm storage (Lin et al, 2006; Joseph et al, 2011). Because morphological abnormalities of

[1]Division of Molecular Pharmacology, Department of Pharmacology, Jichi Medical University, Shimotsuke, Japan   [2]Research Center for Laboratory Animals, Comprehensive Research Facilities for Advanced Medical Science, School of Medicine, Dokkyo Medical University, Mibu, Japan

Correspondence: tutty@jichi.ac.jp

 

epididymal IS have been observed in mice deficient in several genes, including *c-ros* and its ligand Nell2, RNaseIII enzyme Dicer1, and androgen receptor, and these mice were infertile, the proximal parts of the epididymis play a crucial role in male reproductive physiology (Sonnenberg-Riethmacher et al, 1996; Krutskikh et al, 2011; Björkgren et al, 2012; Kiyozumi et al, 2020). The epididymis comprises six cell types: narrow, clear, principal, apical, basal, and halo cells (Cornwall, 2009; Joseph et al, 2011; Shum et al, 2011; Belleannée et al, 2012). The primary cell type, which is present throughout the tubule, is the principal cell type, accounting for 65–80% of the total epithelial cells. Although principal and basal cells are present in all epididymal regions, narrow cells are exclusively located in the IS; clear cells are present in the caput, corpus, and cauda epididymis. Each cell is thought to play a different role because of the significant differences in their gene expression and cellular characteristics (Da Silva et al, 2010). Particularly, narrow and clear cells establish an acidic environment in the epididymal lumen, which is essential for maintaining sperm quiescence during maturation and storage before ejaculation (Shum et al, 2011). Sperms are stored in the cauda epididymis, and after ejaculatory stimulus, they are expelled from the storage through the ductus deferens (Corona et al, 2012; Veening & Coolen, 2014), and then, they swim up the female sexual organs for egg fertilization.

Previous analysis indicated that OTR activates male reproductive function, because spermatogenesis is accelerated in the testes in OTR-overexpressing mice and delayed in OTR-deficient mice (Assinder et al, 2002). However, the phenotype of the V1a-deficient mice remains unknown. In the present study, by analyzing the male reproductive organs in V1a-deficient mice, multiple abnormalities, including morphology of epididymis and altered sperm motility, were detected. Notably, the results also suggested that V1a and OTR exert contrasting effects on spermatozoa. Our findings elucidate the complex mechanisms of sperm maturation and present the V1a receptor as a novel therapeutic target for male infertility.

# Results

### Morphological abnormalities of the epididymis in V1a-deficient mice

A lack of reports on the epididymis characteristics in V1a-deficient mice led us to macroscopically compare the epididymis of WT mice with that of V1a-deficient mice. The epididymis and testes of V1a-deficient mice were much lighter than those of the WT mice ([epididymis] WT: 37.7 ± 1.4 mg, V1a-deficient: 29.0 ± 0.6 mg, $P < 0.001$; [testis] WT: 118.4 ± 3.9 mg, V1a-deficient: 90.4 ± 1.9 mg, $P < 0.001$, n = 8). However, consistent with the findings of a previous study (Aoyagi et al, 2007), our observations revealed that the body weights of the V1a-deficient mice were lower than those of WT (WT: 31.5 ± 0.9 g, V1a-deficient: 24.7 ± 0.8 g, $P < 0.001$, n = 8). Adjustments of the weights of the epididymis and testes according to individual body weights diminished the previously observed significant differences ([epididymis] WT: 1.20 ± 0.04, V1a-deficient: 1.19 ± 0.05, $P > 0.05$; [testes] WT: 3.78 ± 0.14, V1a-deficient: 3.69 ± 0.14, $P > 0.05$, n = 8) (Fig 1A and B).

Other organs were also weighed, and seminal vesicles were significantly larger in the V1a-deficient mice than in the WT mice, whereas the prostate glands were significantly smaller in the V1a-deficient mice than in the WT mice. Similarly in the epididymis, there were no significant differences with respect to the preputial glands ([seminal vesicles] WT: 3.78 ± 0.22, V1a-deficient: 4.42 ± 0.08, $P < 0.01$; [prostate glands] WT: 1.62 ± 0.06, V1a-deficient: 1.26 ± 0.05, $P < 0.005$; [preputial glands] WT: 1.21 ± 0.12, V1a-deficient: 1.25 ± 0.06, $P > 0.05$, n = 10) (Fig S1A).

The longitudinal length of the epididymis in V1a-deficient mice was significantly longer than that in WT mice (WT: 18.6 ± 0.3, V1a-deficient: 19.6 ± 0.5, $P < 0.05$, n = 8) (Fig 1C and D). Furthermore, we analyzed the folding pattern of the epididymal duct because the change in macroscopic morphology may be attributed to malformation of the coiling. The results revealed that the epididymal ducts in the V1a-deficient mice were thinner than those in the WT mice. In the WT mice, the thinnest parts of the epididymis, included in the corpus, normally are comprised of at least three small tubes. However, in the V1a-deficient mice, only one small tube was present in these areas (Figs 1E and S2A and B). The length of the portions of three or fewer tubes was significantly longer in the V1a-deficient mice (WT: 488.9 ± 91.7 $\mu m$, V1a-deficient: 1807.5 ± 294.9 $\mu m$, $P < 0.01$, n = 9–10) (Fig 1F). The overall cross-sectional tissue area of the thinnest part was significantly smaller in V1a-deficient mice than in WT mice (WT: $2.13 \times 10^5 \pm 0.17 \times 10^5 \ \mu m^2$, V1a-deficient: $1.05 \times 10^5 \pm 0.10 \times 10^5 \ \mu m^2$, $P < 0.005$, n = 10–11) (Fig 1G). The area and diameter of the thinnest part of the epididymal duct were also smaller in V1a-deficient mice ([area] WT: $4.53 \times 10^4 \pm 0.35 \times 10^4 \ \mu m^2$, V1a-deficient: $2.94 \times 10^4 \pm 0.31 \times 10^4 \ \mu m^2$, $P < 0.005$; [diameter] WT: 193.5 ± 4.2 $\mu m$, V1a-deficient: 166.4 ± 6.8 $\mu m$, $P < 0.005$, n = 10–11) (Fig 1H and I). The results indicate that deletion of *Avpr1a* led to morphological changes in the epididymis, especially inducing long-axis elongation and central sparseness.

The differences in compartmentalization between the segments of the epididymis can be determined based on the distribution of endogenous $\beta$-galactosidase activity. A previous study reported strong blue color in the IS and corpus epididymis upon $\beta$-galactosidase staining, whereas the caput region was characterized by the absence of a blue color (Sonnenberg-Riethmacher et al, 1996). Our investigation revealed no differences in staining patterns of the epididymis between the V1a-deficient and WT mice (Fig 1J). The result indicated that compartmentalization of the epididymis is normal in V1a-deficient males.

The development of male sexual organs is controlled by androgens such as testosterone (Swain, 2006). To determine whether the weight changes in some male sexual organs observed in the V1a-deficient mice were caused by differences in blood testosterone levels, the concentration of serum testosterone was measured by enzyme immunoassay (EIA). There were no significant differences in serum testosterone levels between the WT and V1a-deficient mice (Fig S1B).

### Distribution of the V1a receptor and OTR in murine tissues

Limited information about the V1a receptor distribution in peripheral tissue led us to perform an in situ hybridization analysis to detect *Avpr1a* transcripts in several organs, including the male

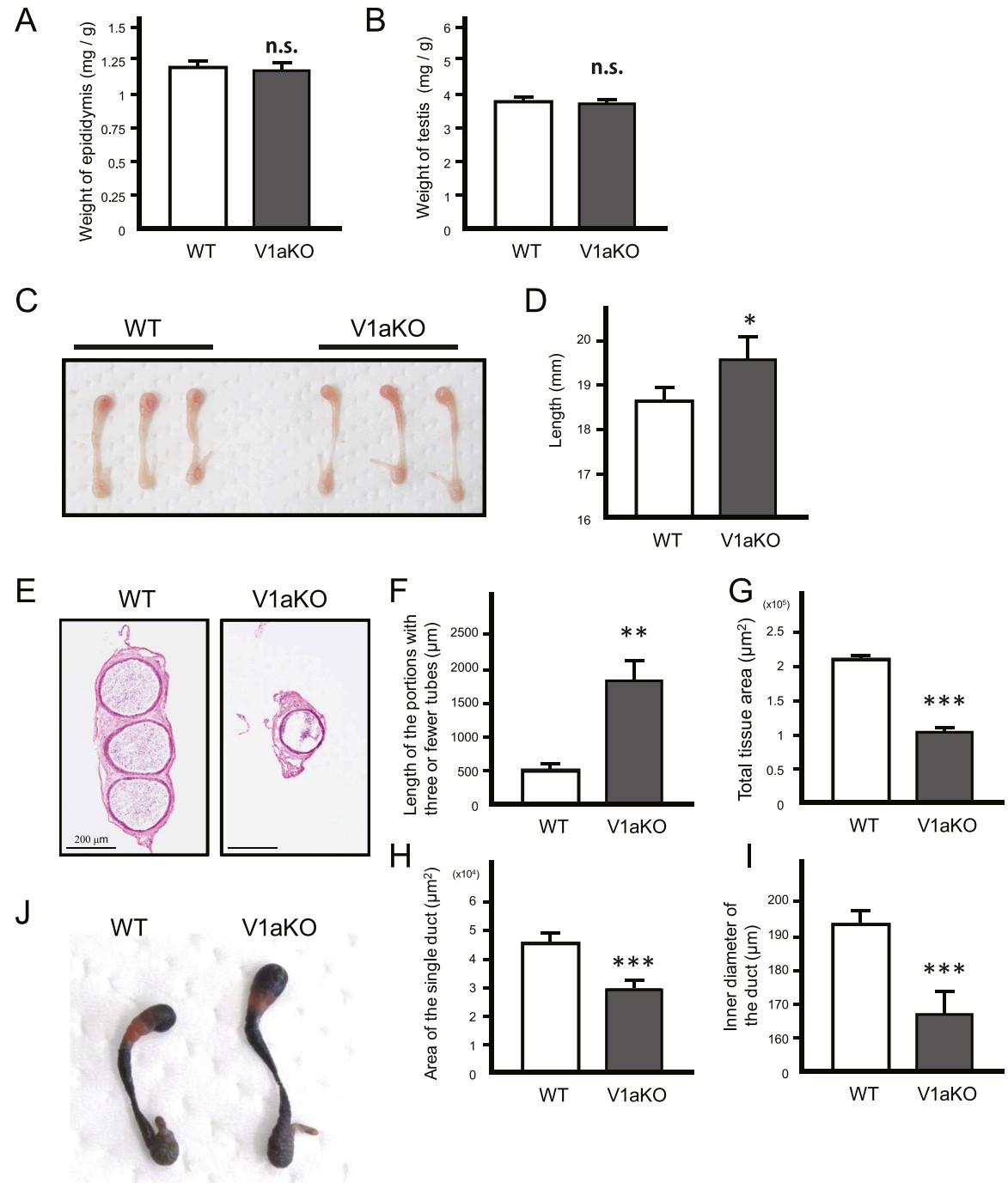

**Figure 1. Morphological abnormalities of the epididymis in V1a-deficient mice.**
**(A, B)** Weights of the epididymis (A) and testes (B) from WT and V1a-deficient (V1aKO) mice were adjusted based on individual body weights (n = 8). However, the differences were not statistically significant. **(C)** Macroscopic morphology of the epididymis in WT and V1aKO mice. **(D)** Longitudinal lengths of the epididymis from WT and V1aKO mice, measured using a ruler (n = 8). **(E)** Thinnest parts of the epididymis at the corpus epididymis of WT and V1aKO mice were compared using various criteria. The images were stained with Mayer's hematoxylin and eosin Y solution. **(F, G, H, I)** Lengths of the portions with three or fewer tubes (F), total tissue area (G), area of the single duct (H), and the inner diameter of the duct (I) (n = 10–11) were compared between WT and V1aKO mice. **(J)** Epididymides of WT and V1aKO were stained using the β-Galactosidase staining procedure. For comparisons between the two groups, we performed Welch's t test. The results are expressed as mean ± SEM, ∗P < 0.05, ∗∗P < 0.01, ∗∗∗P < 0.005, n.s., not significant, versus WT.

genitals. In adult testes, strong *Avpr1a* mRNA signals were detected in the interstitial areas of the seminiferous tubules, which were assumed to be in the Leydig cells (Fig 2A–C). In adult male adrenal glands, although *Avpr1a* mRNA signals were strong in the capsule region, they were scattered and weak in the zona glomerulosa (the main mineralocorticoid-producing area), zona fasciculata (the

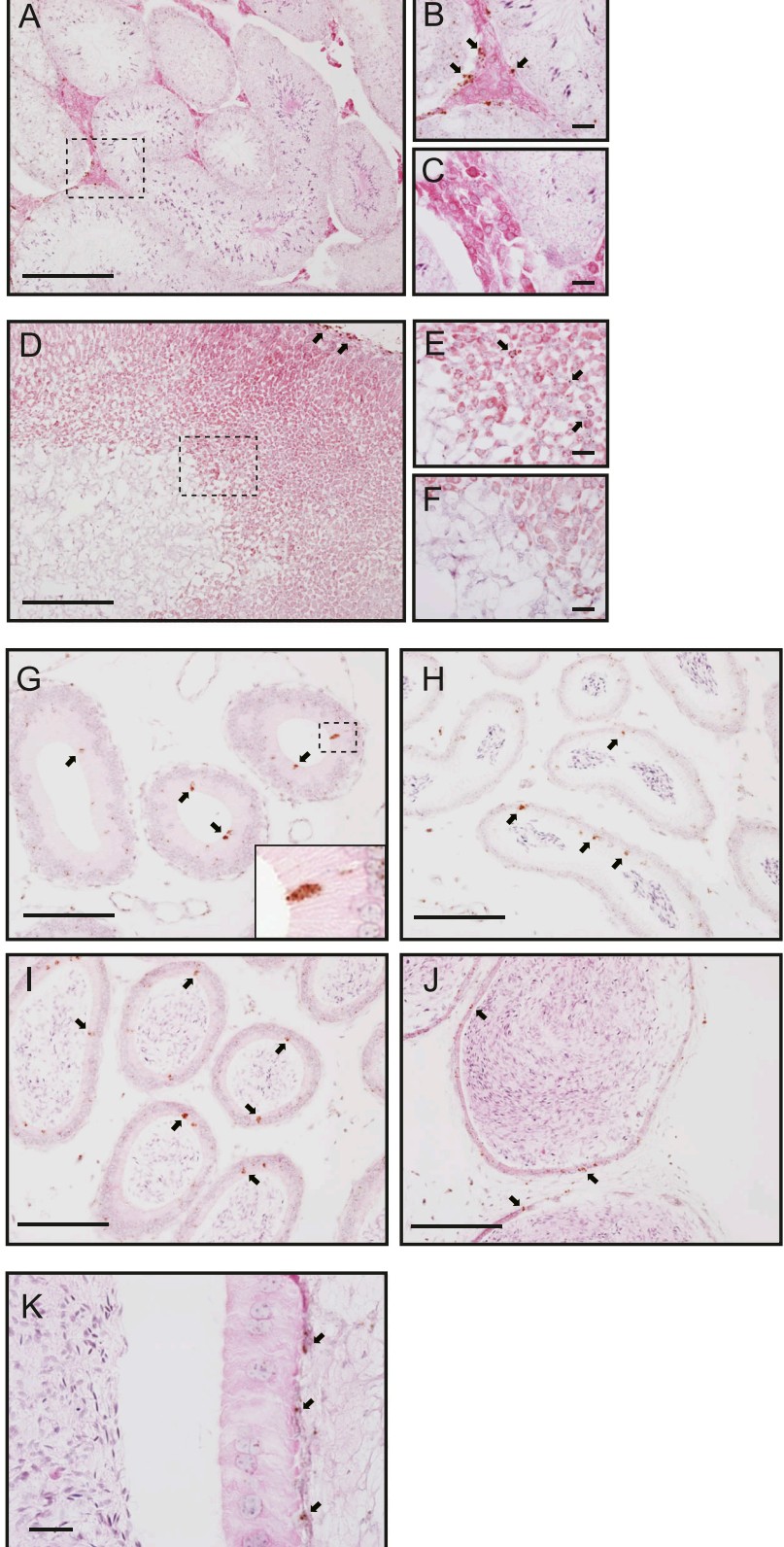

**Figure 2. Distribution of the *Avpr1a* transcripts in murine tissues.**
**(A, B, C, D, E, F, G, H, I, J, K)** In situ hybridization analyses of *Avpr1a* transcripts were performed in the murine testes (A, B, C), adrenal gland (D, E, F), epididymis ((G) initial segment (IS), (H) caput, (I) corpus, (J) cauda), and vas deferens (K). Arrows indicate positive brown signals for the *Avpr1a* transcripts. **(B, E)** Areas surrounded by the dotted lines (A, D), respectively. **(C, F)** The negative controls. **(G)** Inset in (G) shows an area surrounded by dotted lines of (G). Scale bars = 200 μm (A, D), 100 μm (G, H, I, J), and 20 μm (B, C, E, F, K).

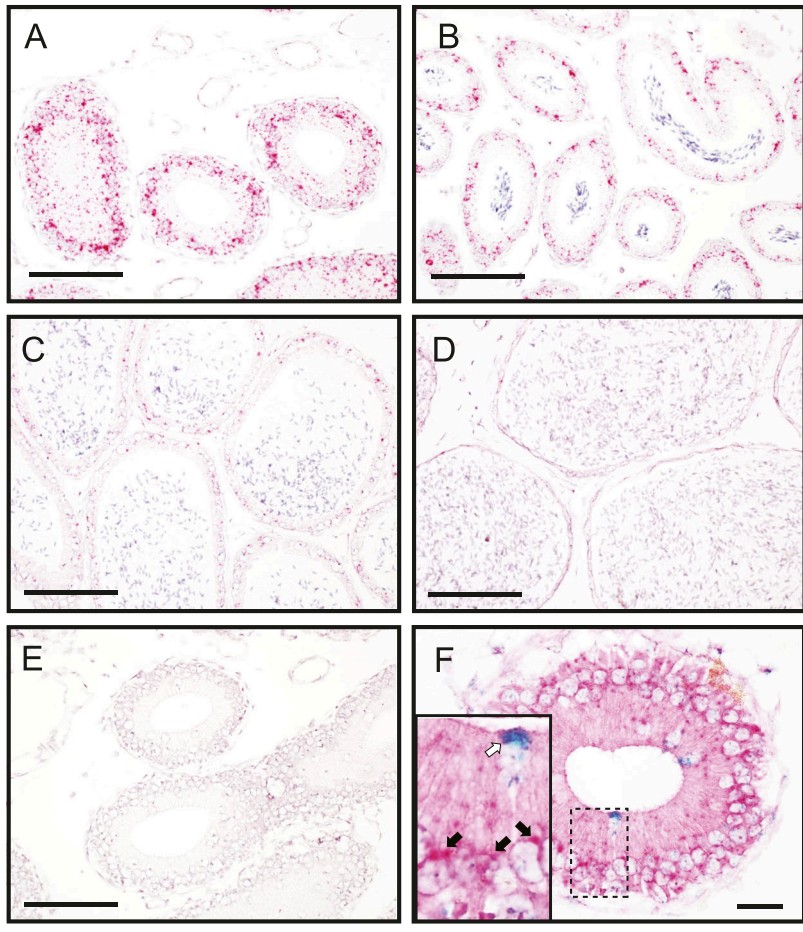

**Figure 3. Distribution of the *Oxtr* transcripts was different from that of *Avpr1a* in murine epididymis.**
**(A, B, C, D, E, F)** In situ hybridization analyses of *Oxtr* transcripts were performed in the murine epididymis ((A, E, F) IS, (B) caput, (C) corpus, (D) cauda). **(E)** represents negative control. **(F)** shows a merged image of double staining with *Avpr1a* transcripts (blue; white arrow in inset) and *Oxtr* transcripts (red; black arrow in inset) in the IS section. Scale bars = 100 *μ*m (A, B, C, D, E) and 20 *μ*m (F).

main glucocorticoid-producing area), adrenal cortex, and adrenal medulla (Fig 2D–F).

We roughly divided the epididymis into four regions: the IS, caput, corpus, and cauda. In the IS, *Avpr1a* signals were strongly detected in dispersed cells in the inner luminal parts of the epididymal duct (Fig 2G). In the caput, corpus, and cauda regions, *Avpr1a* signals were detected adjacent to the cell layer of the epididymal duct (Fig 2H–J). In the vas deferens, strong *Avpr1a* mRNA signals were detected in the basal area of the epithelial cells (Fig 2K). *Avpr1a* transcript was not expressed in the sperm throughout the epididymal duct and vas deferens. The results indicated that *Avpr1a* has characteristic distribution in various tissues, with the exception of sperm.

The mRNA sequence of *Avpr1a* is similar to that of *Oxtr*, which has been suggested to have epididymal functions (Filippi et al, 2002; Studdard et al, 2002; Stadler et al, 2021). Therefore, double staining with probes for *Avpr1a* and *Oxtr* transcripts was performed to reveal differences in expression distribution and minimize chances of cross-detections. The *Oxtr* mRNA signals were broadly detected in the cell layer of all epididymal regions (Fig 3A–E), and the expression signals were stronger in the IS, corroborating the findings of previous studies (Whittington et al, 2001; Filippi et al, 2002). Furthermore, double staining of both mRNAs revealed that the distribution of *Avpr1a* signals was completely different from that of

the *Oxtr*. The result indicated that the *Avpr1a* signals were not attributed to the cross-reactivity of *Oxtr* (Fig 3F).

## V1a receptor signals colocalized with V-ATPase in murine epididymis

Commercially available antibodies to the murine V1a receptor are not suitable for immunohistochemistry because of their low specificity. Therefore, to identify the cell type expressing the V1a receptor in the epididymis, a combined analysis of in situ hybridization and immunohistochemistry was performed in the present study. Based on the distribution pattern of *Avpr1a* signals, we deduced that the *Avpr1a*-expressing cells were narrow/clear cells, and this was confirmed by double staining of the samples with V-ATPase, a marker for narrow/clear cells (Miller et al, 2005; Da Silva et al, 2007b), and the *Avpr1a* transcripts. *Avpr1a* signals were exclusively detected in the perinuclear zone of the cells, which coexpressed the V-ATPase signals in the luminal region (Figs 4 and S3). No apparent difference in histological appearance and distribution of V-ATPase in any compartment of the epididymis between the V1a-deficient mice and WT mice led us to hypothesize that the non-functionality of the V1a receptor does not influence the V-ATPase expression or the distribution of the narrow/clear cells (Figs S4 and S5). Narrow/clear cells participate in the

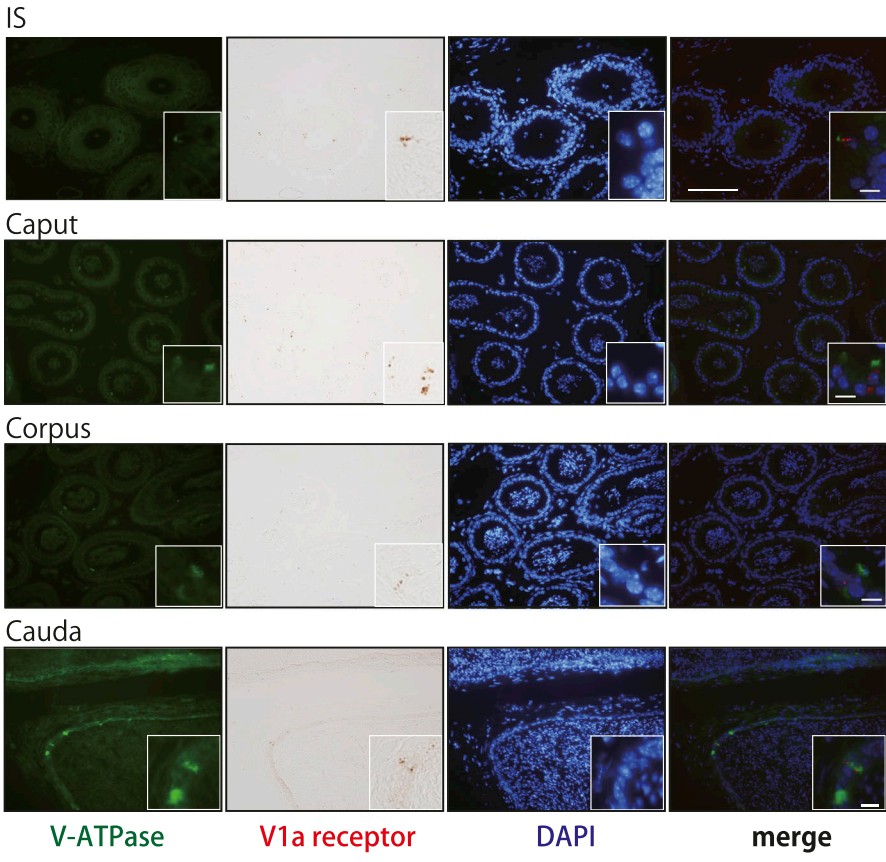

IS

Caput

Corpus

Cauda

**V-ATPase**    **V1a receptor**    **DAPI**    **merge**

**Figure 4.  *Avpr1a* signals colocalized with V-ATPase in murine epididymis.**
In the merged images, red represents *Avpr1a* transcripts (pseudocolor) and green represents V-ATPase protein. Blue indicates the nuclei. Scale bars = 100 and 10 μm (inset).

maintenance of the luminal environment of the epididymis (Da Silva et al, 2007a; Breton & Brown, 2013). Therefore, our results suggested that the V1a receptor regulates the luminal condition of the epididymis by controlling narrow/clear cell activation.

### Sperm motility and activity were up-regulated in V1a-deficient mice

Subsequently, we analyzed motility kinetics to investigate sperm motility and activity in V1a-deficient mice (Fig 5A). Straight-line velocity (VSL) and beat–cross frequency (BCF) of spermatozoa in the V1a-deficient mice were significantly higher than those in the WT mice. In the linearity (LIN) analysis, the spermatozoa from V1a-deficient mice showed low linearity for the first time; however, 2 h after the incubation, it showed significantly higher linearity than the WT. Other factors, including curvilinear velocity (VCL), average path velocity (VAP), straightness (STR), amplitude of lateral head displacement (ALH), and wobbler coefficient (WOB), showed little difference in the genetic background. Furthermore, significantly long-lasting motility and high probability of hyperactivation were observed in V1a-deficient mouse spermatozoa compared with those in the WT spermatozoa (Fig 5B), with no major abnormalities in sperm morphology. The results suggested that spermatozoa from the V1a-deficient mice may be more active than those from WT mice.

### Hyperactivation was inhibited in atosiban-treated mice

The activation of sperm motility and hyperactivation of spermatozoa were observed in the V1a-deficient mice. Next, we analyzed the effect of OTR defects on sperm motility and hyperactivation. Because Assinder and coworkers have already reported abnormal maturation of sperm in OT-deficient mice (Assinder et al, 2002), we used the original method with the antagonist atosiban, which is not expected to cause growth inhibition of the sperm, as observed in OT-deficient mice.

Upon atosiban administration, no difference in motility was observed between the vehicle- and atosiban-treated spermatozoa (Fig 6). In contrast, sperm hyperactivation was significantly attenuated by atosiban treatment. The results suggested that OTR inhibition in adults disturbs normal sperm maturation.

### AVP stimulation increased the extracellular pH mediated by the V1a receptor

V-ATPase on the plasma membrane exports protons to the extracellular space, changing the intracellular pH to alkaline and the extracellular pH to acidic. In addition, it is considered that low pH in epididymal lumen keeps sperm quiescent during their storage and a pH increase in oviduct fluid after ejaculation initiates hyperactivation (Blomqvist et al, 2006; Suarez, 2008; Shum et al, 2011).

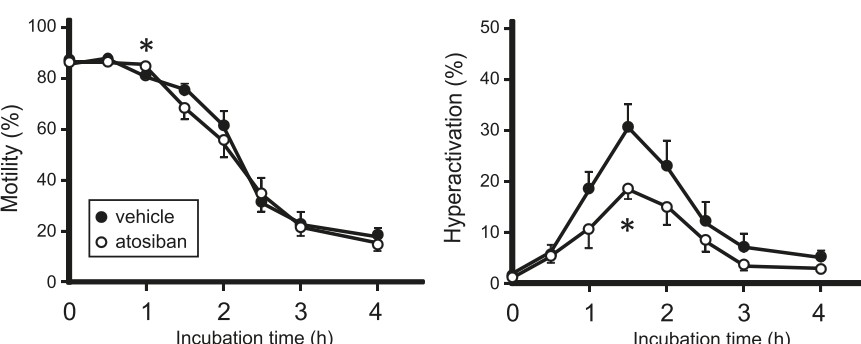

**Figure 5. Increased sperm activity in V1a-deficient mice (white) than in WT (black) mice.**
**(A)** Straight-line velocity, curvilinear velocity, average path velocity, linearity, straightness, amplitude of lateral head displacement, beat–cross frequency, and wobbler coefficient were recorded and calculated using a computer (n = 8). **(B)** Persistent motility and hyperactivation were measured manually (n = 8). For comparisons between the two groups, we performed Welch's t test. The results are expressed as mean ± SEM, ∗P < 0.05, ∗∗P < 0.01, ∗∗∗P < 0.005, versus WT values.

**Figure 6. Hyperactivation of the spermatozoa was up-regulated in 14-d atosiban-injected mice (white) compared with control (black).**
Persistent motility and hyperactivation were manually measured. For comparisons between the two groups, we performed Welch's t test. The results are presented as mean ± SEM, ∗P < 0.05, versus the vehicle group (n = 8).

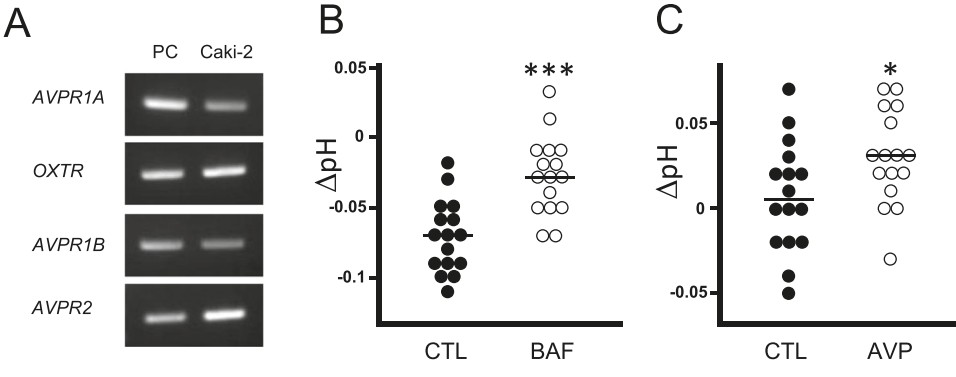

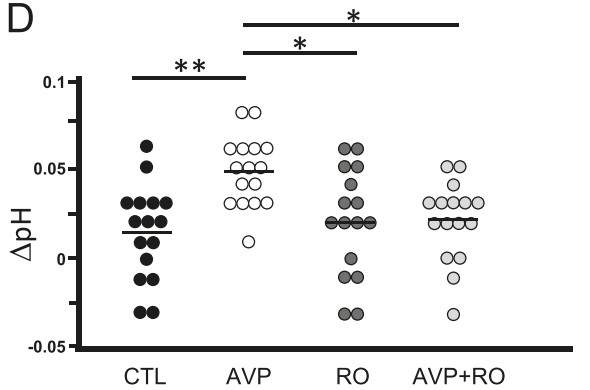

**Figure 7. AVP stimulation increased the extracellular pH, and V1a receptor antagonist blocked it.**
**(A)** Gene expression levels of receptor subtypes of AVP and oxytocin in human renal carcinoma Caki-2 cells. Human liver, brain, and kidney were used as a positive control for *AVPR1A*, *AVPR1B* and *OXTR*, and *AVPR2*, respectively. **(B)** pH changes from the background buffer were plotted in the graph. **(B, C)** 1 $\mu$M bafilomycin A1 (BAF) or 1 $\mu$M AVP treatment has been compared with the vehicle (CTL). For comparisons between the two groups, we performed Welch's *t* test. The bars represent median, *$P < 0.05$, ***$P < 0.005$, versus the CTL group (n = 16). **(D)** pH changes induced by AVP with or without V1a antagonist RO5028442 (RO) were plotted. For comparisons between multiple groups, Bartlett's test followed by one-way ANOVA and Tukey's test were combined. The bars represent mean, *$P < 0.05$, **$P < 0.01$, versus the CTL group (n = 16).

Therefore, strict pH controls of reproductive tissues are critical for fertilization. Our results indicate that *Avpr1a* is coexpressed in the narrow/clear cells of the epididymis in which V-ATPase is expressed, indicating that the V1a receptor may also have the ability to change pH. We subsequently examined how the AVP-V1a receptor activation affects pH. Unfortunately, narrow/clear cells of the epididymis are scattered, making it difficult to use them to investigate changes in pH. Therefore, the pH measurement experiment was performed in the cell line of the kidney, which is close in origin to the epididymis (Hinton & Turner, 1988; Hess, 2002). The use of cell line with uniform expression characteristics has the advantage of making it easier to detect changes in pH during agonist stimulation. In this report, we selected Caki-2 cells, human renal carcinoma cells, which express V-ATPase and have a proven experimental procedure for pH measurement (Al-Bataineh et al, 2016).

We first analyzed the expression of the receptor subtypes of AVP and OT in Caki-2 cells, and all receptor subtypes could be detected (Fig 7A). To investigate whether V-ATPase functions in Caki-2 cells, we measured pH changes induced by proton export into the extracellular assay solution in the presence of bafilomycin A1, an inhibitor of V-ATPase. The increase in extracellular pH indicated that bafilomycin A1 could inhibit V-ATPase function and that proton transport had stopped (Fig 7B). When the pH in the extracellular assay solution was measured after AVP treatment in a similar way, the extracellular pH also increased (Fig 7C). The results indicated that AVP stimulation could have inhibited proton release. Because all AVP receptor subtypes are expressed in Caki-2 cells, it is unclear

which receptors are activated to change pH after AVP stimulation. To demonstrate that the change was V1a receptor–mediated, we used the specific antagonist RO5028442 (Ratni et al, 2015). As a result, the simultaneous addition of RO5028442 suppressed the AVP-induced increase in pH (Fig 7D). In other words, the AVP-induced pH increase was mediated by the V1a receptor.

## Discussion

Previous reports have demonstrated that morphological abnormalities of the epididymis are often severe in several deficient mice, resulting in infertility. For example, deletion mouse models of lumicrine factor Nell2 or its receptor, c-ros, exhibited abnormal IS differentiation (Sonnenberg-Riethmacher et al, 1996; Kiyozumi et al, 2020). In addition, the RNaseIII enzyme Dicer1-deficient mice and epididymis-specific conditional androgen receptor–deficient mice also had an abnormality in IS and were infertile (Krutskikh et al, 2011; Björkgren et al, 2012). However, to the best of our knowledge, no study has demonstrated the characteristic abnormalities of the different folding numbers observed in the corpus epididymis of the V1a-deficient mice. Our results suggest that one of the reasons for the altered male reproductive physiology in the V1a-deficient mice is macroscopic structural changes in the epididymis. Several factors, including testosterone, inhibin, and the autosomal dominant polycystic kidney disease gene *pkd1*, are considered important in the coiling of the epididymal duct (Welsh et al, 2006; Joseph et al, 2009; Nie & Arend, 2013). However, there was no significant

difference in serum testosterone levels between the WT and V1a-deficient mice.

In general, V1a receptors produce contractile forces when ligands bind and activate them, as exemplified by the contraction of blood vessels and the uterus. The system is also considered to be active during epididymal differentiation. Alternatively, under normal conditions, cells divide and epididymal tubules elongate during differentiation. However, such tubules are unable to expand because of spatial limitations and are forced to form a tightly coiled structure. In V1a-deficient mice, we hypothesized that the longitudinal pressure required to suppress the spread of the epididymal duct decreased. Therefore, the epididymal tubules could occupy more space, reducing the amount of folding in the corpus epididymis and increasing the macroscopic length, as observed in our analysis. Further studies investigating the differentiation and growth processes of the epididymis are required to validate this hypothesis.

In the present study, sperm motility was activated in the V1a-deficient mice; however, litter size did not increase in pregnancies involving V1a-deficient male mice (Tsuchiya et al, 2020). The observation indicated a probability of an upper limit for the number of fertilized eggs that can be produced and implanted, despite improvement in sperm motility in the deficient mice. However, we propose that if fertility is compromised by other paternal factors, blocking the V1a receptor may ameliorate pregnancy defects.

Our study revealed that *Avpr1a* mRNA signals were scattered in the zona glomerulosa and zona fasciculate of adrenal cortex, and adrenal medulla. The results corroborated previous findings showing the involvement of the V1a receptor in the physiological activity of mineralocorticoid secretion from the zona glomerulosa cells and the corticosterone response to adrenocorticotropic hormone (Guillon et al, 1995; Grazzini et al, 1999; Koshimizu et al, 2006; Birumachi et al, 2007). In addition to the production of glucocorticoids and mineralocorticoids, the adrenal glands also produce adrenergic androgens such as dehydroepiandrosterone and dehydroepiandrosterone sulfate in primates (Rainey et al, 2002). However, there are species differences in the tissue constructions of the adrenal glands, and the secretions of adrenergic androgens are absent in rodents (Pihlajoki et al, 2015). In fact, we could not observe the clear zona reticularis, which was the main layer of the androgen production in the murine adrenal gland. In the future, we would like to compare the present results with the expression in androgen-producing species and clarify the differences in the distribution of V1a receptors between the sexes.

The OT and AVP acting on the epididymis and testes are not of hypothalamic origin but are produced by the organs upon which the hormones act, because the circulating hormone levels in the blood do not correspond to the concentration of hormones in the target organs such as the epididymis and testes (Kasson et al, 1986; Assinder et al, 2000). Therefore, the ligands of the V1a receptor in the present study were OT or AVP produced in the testes or epididymis, respectively. In fact, the expression of AVP or OT in the testis or epididymis has been observed in various species (Knickerbocker et al, 1988; Fillion et al, 1993; Einspanier & Ivell, 1997; Assinder et al, 2000). However, the extensive species-specific differences in their expressions warrant further research to understand how they are used differently.

Atosiban is a peptide analogous to the hypophyseal hormones AVP and OT and is used in Europe as a tocolytic agent for patients with imminent premature birth (Craciunas et al, 2021). Species differences in the V1a and OTR sequences were not less pronounced. However, whereas atosiban is a competitive V1a/OTR antagonist in humans and rats, it can only inhibit ligand binding to murine OTR and not to the V1a receptor (Akerlund et al, 1999; Cirillo et al, 2003; Busnelli et al, 2013). Therefore, atosiban acts as a specific OTR antagonist in mice. A previous study by Pierzynski et al, which analyzed the effects of atosiban on motility and hyperactivation of human sperm, concluded that it has no effect on the sperm (Pierzynski et al, 2007). However, in our study, atosiban inhibited sperm hyperactivation. This contradicting result could stem from a fundamentally different experimental system. In a previous study, sperms were incubated in vitro in a medium containing atosiban for 24 h. However, in the present study, atosiban was administered systemically (intraperitoneally) for 14 d, which was comparable to the sperm storage period in the murine epididymis (10–12 d) (Sullivan & Mieusset, 2016). The results suggest that atosiban does not directly cause an adverse effect on sperm, but indirectly impairs sperm function by altering the environment of the epididymis. In fact, neither OTR nor V1a mRNA expression was observed in the spermatozoa in our in situ hybridization analysis; therefore, it is plausible that no direct action was observed in the previous study. Therefore, OTR may have some functions in the sperm maturation process rather than after sperm ejaculation. It has also been reported that OT does not directly affect sperm motility (Berndtson & Igboeli, 1988; Walch et al, 2001; Byrne et al, 2003). In contrast, OT has been reported to increase the number of ejaculated spermatozoa in several species (Fjellström et al, 1968; Knight & Lindsay, 1970; Agmo et al, 1978; Nicholson et al, 1999; Filippi et al, 2002). Such changes may be mediated by the activation of OTR in the testis or epididymis, which results in sperm accumulation in the epididymis or increased contractility of the epididymal tubules and vas deferens (Hib, 1974; Knight, 1974; Jaakkola & Talo, 1981; Studdard et al, 2002). Our results suggested that maturation abnormalities in spermatozoa occurred when male reproductive organs were exposed to chemicals that interrupt the function of OTR, or males suffer from diseases that cause abnormality in OT function. However, further research is required to determine whether atosiban affects the sperm production or maturation and storage stages in the epididymis of the testes.

Numerous studies have reported the distribution of OTR in male reproductive tissues (Einspanier & Ivell, 1997; Frayne & Nicholson, 1998; Whittington et al, 2001; Filippi et al, 2002), highlighting the presence of OTR in the muscle layer of the epididymis, corroborating our results. However, very few reports have demonstrated the distribution of V1a receptors in male reproductive tissues. Previous studies have indicated that V1a receptors are distributed in Leydig cells (Phillips et al, 1990), and AVP controls testosterone production in these cells in rats (Stadler et al, 2020). However, no previous studies have investigated V1a expression patterns in the cells of the epididymis. The lack of sensitive and specific antibodies to the V1a receptor protein, low protein expression, and sequence homology between OT and AVP receptors may account for the challenge in detecting V1a receptor expression. In this experiment, we showed the epididymal distribution of specific OTR and V1a

receptors using the RNAscope technique established in previous reports (Tsuchiya et al, 2020). As shown in Figs 2 and 3, the expression distribution of the OTR and V1a receptor in the epididymis was completely different. These results suggest that the OTR and V1a receptor exert different regulatory effects on male reproductive functions in different epididymal cells.

Interestingly, our analysis revealed that V1a receptors were distributed in the narrow/clear cells of the epididymis. The V-ATPase expression suggested that narrow/clear cells regulate the pH in the epididymal duct, which is essential for proper germ cell maturation and maintenance of sperm in a quiescent state (Blomqvist et al, 2006; Shum et al, 2011). To summarize, the expression of V1a receptors in the narrow/clear cells suggested that the V1a receptor also regulates pH in the epididymal tubules after ligand binding. A similar case has already been analyzed in the kidney, where V1a coexists with V-ATPase to regulate the pH of the tubules (Giesecke et al, 2019). These changes in the environment within the epididymal tubules may be responsible for the enhanced sperm motility observed in the V1a-deficient mice.

Our results highlighted the physiological function of the V1a receptor in inhibiting sperm maturation, whereas OTR promotes sperm maturation. Sperms stored in the cauda epididymis strictly remain in a sedated state. However, after ejaculation, they immediately become active and migrate up the vagina toward the oocytes, a process in which the sperm activity must be tightly controlled for the normal sperm function. We hypothesized a probable mechanism, wherein V1a and OTR may be responsible for exerting such complementary contradicting functions. OTR is strongly and widely expressed in male reproductive tissues, including the epididymis. Our results showed that, despite being analogous receptors, the action of the V1a receptor is completely different from that of OTR, and is also important. This implies that the use of the V1a receptor and OTR is very strictly regulated. These results indicated that not only the OTR but also the V1a receptor is important in future analyses of male reproductive function and development. The findings also suggest that a highly specific V1a antagonist could improve the function of sperm with weakened motility because of pathology or aging. In addition, therapeutics that inhibit the functions of pituitary hormone receptors can cause reproductive side effects.

It has been reported that the expressions and distributions of the V1a receptor and OTR in the central nervous system differ among species (Freeman & Young, 2016). The epididymis is also known to have structural differences among species (Sullivan & Mieusset, 2016). For example, IS is absent in the human epididymis, and c-ros, an IS marker in mice, showed a broad distribution in epididymis (Légaré & Sullivan, 2004). However, there is a similar distribution pattern of clear cells between rodents and humans, and V-ATPase is thought to function in those cells of both species (Sullivan et al, 2019). Future studies using human tissues are required to determine whether the V1a receptor is also expressed in human epididymal clear cells and influences sperm function.

Our experimental results showed that AVP stimulation increased pH and the effect was abolished by V1a antagonist RO5028442, which exhibits little binding to OTR (Ratni et al, 2015). The results suggest that AVP stimulation suppresses proton release of V-ATPase mediated by the V1a receptor. In other words, the activation of sperm motility observed in V1a-deficient mice appears to result from the cancellation of the inhibitory function of V-ATPase by the V1a receptor within the epididymis. Activation of V-ATPase and subsequent acidification of the tubules are conditions that induce sperm quiescence. Maintaining strict dormancy may be important for sustained motility and frequency of sperm hyperactivation observed here.

However, whether this change was directly exerted by the association between V-ATPase and V1a receptor or indirectly regulated through changes in intracellular signaling requires further detailed analysis. Little is known about the interaction between G protein–coupled receptors and V-ATPase. It is widely known that activation of the Gs protein increases intracellular cAMP and subsequent activation of PKA, whereas activation of the Gq protein increases intracellular calcium concentrations. In addition, it is generally well known that PKA activates V-ATPase, but there are only a few reports on the relationship between calcium ion and V-ATPase (Crider & Xie, 2003; Hallows et al, 2009; Al-bataineh et al, 2014; McGuire et al, 2016; Zaidman et al, 2020). Although the V2 receptor couples with the Gs protein, the other vasopressin receptor subtypes, including the V1a receptor, couple with the Gq protein. Therefore, assuming that indirect signaling is involved, calcium increase is a key factor, but further analysis is required to clarify the details. Our results indicate a novel potential interaction between G protein–coupled receptors and V-ATPase.

Our presentation of the V1a receptor as a candidate therapeutic target for sperm motility disorder and the identification of its regulatory role in sperm maturation, which significantly differs from that of OTR, warrants further analyses of the mechanisms of sperm maturation and development of treatment methods for infertility.

# Materials and Methods

### Animals

*Avpr1a*-deficient mice (Avpr1a$^{tm1Gzt}$/Avpr1a$^{tm1Gzt}$, denoted here as V1a-deficient mice, 25–30 g) were first described by Koshimizu et al (2006). V1a-deficient mice and control WT mice were maintained on a hybrid 129/Sv and C57BL/6J background and bred in our laboratory, as previously described (Tsuchiya et al, 2020). The animals were housed in a light (12 h on/off)- and temperature-controlled (23 ± 2°C) facility, with food and water available ad libitum. All mouse-related experiments were approved by the Animal Care and Use Committee of Jichi Medical University. At the time of sample collection, all animals were either subjected to deep anesthesia with isoflurane (Wako Pure Chemical Industries) or euthanized by cervical dislocation, and efforts were made to minimize suffering.

### Qualitative analysis of the epididymal tissues

The seminal vesicle, preputial gland, prostate gland, testes, and epididymis were collected from male mice aged 13–22 wk, and the weights of those tissues were recorded. Sampled epididymides were immediately embedded in Tissue-Tek OCT compound (Sakura), frozen, and sliced into 12-$\mu$m sections using a cryostat

(Leica Microsystems). Serial sections of the thinnest part of the corpus epididymides were stained with Mayer's hematoxylin and eosin Y (HE; Wako Pure Chemical Industries) and examined under a light microscope (AX80; Olympus).

### β-Galactosidase staining

β-Galactosidase staining was performed as previously described (Björkgren et al, 2012). The tissues were fixed in 0.2% glutaraldehyde, 2 mM $MgCl_2$, and 5 mM ethylene glycol-bis(β-aminoethyl ether)-N,N,N',N'-tetraacetic acid in PBS for 30 min at room temperature (RT: usually 23 ± 2°C). After washing overnight, the tissues were stained for 2 h at 37°C in 2 mM $MgCl_2$, 0.01% Na-deoxycholate, 0.02% NP-40, 5 mM $K_4Fe(CN)_6$, 5 mM $K_3Fe(CN)_6$, and 1 mg/ml X-gal in PBS. After incubation, the epididymides were washed with PBS. The staining was repeated for the epididymis of five WT and five V1a receptor–deficient mice, all with similar results.

### Testosterone measurement by EIA

To determine the concentration of testosterone in adult WT and V1a-deficient male mice, serum from each mouse was collected under isoflurane anesthesia and purified by ether extraction. Quantitative analysis of testosterone was performed in duplicate using an EIA kit (Cayman). The signals on the reaction plate were read at 405 nm using an ARVO MX spectrophotometer (PerkinElmer).

### In situ hybridization analysis

Adult male WT and V1a-deficient mice were euthanized and fixed by transcardial perfusion using 4% paraformaldehyde solution (Nacalai Tesque). Epididymides were collected and incubated for 16 h in 4% paraformaldehyde at RT. The next day, they were embedded in paraffin blocks and sliced into 5-μm sections. RNA in situ hybridization was performed using an RNAscope 2.5 HD Detection Kit according to the manufacturer's protocol (Advanced Cell Diagnostics). The sections were hybridized with probes against *Avpr1a* (no. 418061) or *Oxtr* (no. 402651-C2) mRNA or with a negative control probe (no. 310043). The tissues were counterstained with Mayer's hematoxylin and eosin Y solution and observed under a light microscope (AX80). The manufacturer presented the signal color for *Avpr1a* as green in the double-staining experiment, but we describe it as blue in this report. All experiments were repeated on tissues from three different male mice, with the same results.

### Immunohistochemistry analysis

Epididymal sections (5 μm thick) of adult male WT and V1a receptor–deficient mice were prepared in the same way as for the in situ hybridization analysis. The slides were incubated in incubation buffer (1% fetal bovine serum and 0.3% Triton X-100 in PBS) containing anti-V-ATPase B1 and B2 subunit antibodies (ab200839, 1:1,000; Abcam) overnight at RT. After washing with 0.3% Triton X-100–containing PBS, tissue slices were incubated in the incubation buffer containing the secondary antibody

(Alexa Fluor 488–conjugated anti-rabbit IgG [A-11034], 1:200; Thermo Fisher Scientific) for 1 h at RT. DAPI (Dojindo) was used as the nuclear marker. Sections were then mounted using ProLong Gold antifade reagent (Thermo Fisher Scientific), and images were obtained under a fluorescent microscope (AX80). To demonstrate the specificity of the primary antibody, experiments were performed in which epididymal sections were incubated with incubation buffer alone instead of the primary antibody as a negative control.

For double staining of the V1a receptor mRNA and V-ATPase protein, in situ hybridization analysis was performed following the above-mentioned protocol. Upon detecting the signals of the *Avpr1a* mRNA, immunohistochemistry staining for the V-ATPase protein was performed. All immunohistochemistry experiments were repeated on tissues from three different male mice, with the same results.

### Motility kinetics

Motility kinetics was evaluated using the Sperm Motility Analysis System (SMAS) for animals (Ver. 3.18) with the loaded parameter file mouse_BM10×_640 nm_Bright59_150fps-shutter200. ini (Ditect) (Sugiyama et al, 2019). Spermatozoa were obtained from the cauda epididymis of sexually mature male WT or V1a-deficient mice (12–18 wk old). The number of animals was adjusted to be as equal as possible within the comparison group, using males born from three or more mothers. Modified Tyrode's albumin lactate pyruvate (mTALP) medium, containing 101.02 mM NaCl, 2.68 mM KCl, 2 mM $CaCl_2$-$2H_2O$, 1.5 mM $MgCl_2$-$6H_2O$, 360 μM $NaH_2PO_4$-$2H_2O$, 35.70 mM $NaHCO_3$, 4.5 mM D-glucose, 90 μM sodium pyruvate, 9 mM sodium lactate, 500 μM hypotaurine, 50 μM (-)epinephrine, 200 μM sodium taurocholate, 5.26 μM sodium metabisulfite, 0.05% (wt/vol) streptomycin sulfate, 0.05% (wt/vol) potassium penicillin G, and 15 mg/ml BSA (pH 7.4 at 37°C under 5% (vol/vol) $CO_2$ in air), was used as a capacitation medium (Maleszewski et al, 1995). A drop (~3 μl) of cauda epididymal spermatozoa was placed in a culture dish (35 mm diameter) with 3 ml of the mTALP medium, followed by incubation at 37°C for 5 min to allow the spermatozoa to swim up. The mTALP medium containing motile spermatozoa was placed in a new culture dish and incubated for 4 h at 37°C under 5% $CO_2$ concentration to allow hyperactivation. The suspension containing motile spermatozoa (20 μl) was transferred to an observation chamber (0.1 mm deep, 18 mm wide, and 18 mm long) made of mending tape attached to the glass slide in two parallel strips and then covered with coverslips. The movement of spermatozoa was recorded for 1 s on the hard disk drive of the SMAS using a high-speed digital camera (HAS-L2; Ditect) attached to a microscope (ECLIPSE E2000; Nikon) with phase-contrast illumination, a 650-nm band-pass filter, and a warm plate (MP10DM; Kitazato Corp.). The SMAS analyzed 150 consecutive images obtained from a single field at 10× magnification with negative phase contrast. VSL (μm/s), VCL (μm/s), VAP (μm/s), linearity (LIN; defined as VSL/VCL), straightness (STR; defined as VSL/VAP), amplitude of lateral head displacement (ALH, μm), and beat–cross frequency (BCF, Hz) were automatically calculated by the SMAS; the wobbler coefficient (WOB; defined as VAP/VCL) was calculated manually. For each SMAS experiment, ≥300 spermatozoa were detected. Only motile spermatozoa that were significantly different were analyzed.

## Measurement of the percentage of motile and hyperactivated spermatozoa

Analyses of motile and hyperactivated spermatozoa were performed according to a previously described method (Sugiyama et al, 2019), with some modifications. Spermatozoa were obtained from the cauda epididymis of sexually mature male WT or V1a-deficient mice (16–22 wk old). A drop (~3 $\mu$l) of cauda epididymal spermatozoa was placed in a culture dish with 3 ml of the mTALP medium, followed by incubation at 37°C for 5 min to allow the spermatozoa to swim up. The mTALP medium containing motile spermatozoa was placed in a new culture dish and incubated for 4 or 6 h at 37°C under 5% $CO_2$ to allow hyperactivation to occur. Motile spermatozoa were recorded on a DVD recorder (RDR-HX50; Sony Corp.) using a CCD camera (Progressive 3CCD, Sony) attached to a microscope (IX70; Olympus Corp.) with phase-contrast illumination in a small $CO_2$ incubator (MI-IBC; Olympus). Each observation was performed at 37°C, recorded for 1 min, and analyzed by manually counting the number of total spermatozoa, motile spermatozoa, and hyperactivated spermatozoa in four different fields of observation. The analyses were performed in a blinded manner. Sperms exhibiting beating flagellar movement were defined as motile sperm, and motile sperms exhibiting asymmetric and whiplash-like flagellar movements and a circular and/or octagonal swimming locus were defined as hyperactivated sperm. The percentage of motile spermatozoa was defined as the ratio of motile spermatozoa to the total spermatozoa multiplied by 100 (%). The percentage of hyperactivated spermatozoa was defined as the ratio of hyperactivated spermatozoa to total spermatozoa multiplied by 100 (%).

## Atosiban treatment

Atosiban was purchased from ProSpec-Tany TechnoGene. The number of animals was adjusted to be as equal as possible within the comparison group, using males born from three or more mothers. In addition, care was taken to ensure that littermates were assigned to both the control and atosiban-treated groups. Male WT mice (16–19 wk old) were injected daily with 1 mg/kg atosiban peritoneally for 14 d. 24 h after the last injection, mice were euthanized by cervical dislocation, and their cauda epididymides were collected and placed in mineral oil at 4°C. The obtained spermatozoa were analyzed for motility and hyperactivation according to the method mentioned above.

## Measurements of changes in extracellular pH

Caki-2 human kidney carcinoma cells were obtained from the American Type Culture Collection and cultured in McCoy's 5a medium (Thermo Fisher Scientific) supplemented with 10% fetal bovine serum (Thermo Fisher Scientific), 100 U/ml penicillin (Thermo Fisher Scientific), and 100 mg/ml streptomycin (Thermo Fisher Scientific) according to the supplier's recommendation. The pH measurements were made according to a previous report (Al-Bataineh et al, 2016). First, Caki-2 cells were seeded onto 24-well plates at 80–95% confluence and maintained in culture media at 37°C with 5% $CO_2$ on an air-ventilated humidified incubator. 24 h later, the medium was replaced with a Na⁺-free, low-buffering capacity solution for pH measurement (135 mM N-methyl-D-glucamine, 5 mM KCl, 2 mM $CaCl_2$, 1.2 mM $MgSO_4$, 5.5 mM D-glucose, 6 mM L-alanine, 4 mM lactic acid, and 1 mM Hepes, titrated to pH 7.43 using HCl), containing reagents. After the optimal incubation time (20 min; 1 $\mu$M bafilomycin A1 [Cayman], and 1 h; 1 $\mu$M AVP [Peptide Institute] and V1a antagonist 10 nM RO5028442 [Cayman] [Ratni et al, 2015]), the buffer solution was applied to the pH meter (LAQUAtwin-pH-22B; HORIBA Advanced Techno). The results are presented as changes in the control (DMSO) or reagent groups minus the background value (buffer solution only).

## Gene expression analysis

Total RNA was extracted from confluent Caki-2 cells with TRIzol reagent (Thermo Fisher Scientific). The total RNA (5 $\mu$g) was reverse-transcribed in the presence of oligo-dT primers with SuperScript III First-Strand Synthesis System (Thermo Fisher Scientific) according to the manufacturer's protocols. PCRs were performed using a C1000 Touch Thermal Cycler (Bio-Rad) using PCR enzyme (TaKaRa Ex Taq; TaKaRa Bio) and specific primers (listed in Table 1). Total RNA samples from human liver, brain, and kidney (BioChain) were used as a positive control (PC) for *AVPR1A*, *OXTR* and *AVPR1B*, and *AVPR2*, respectively.

## Data analysis

For comparisons between the two groups, we performed Welch's *t* test. Statistical significance was set at $P < 0.05$. The data are summarized in the corresponding figures using mean or median values. For comparisons between multiple groups with comparable variances, Bartlett's test was performed first, followed by one-way ANOVA. If we could show that there was a significant difference in ANOVA, the parametric data were analyzed using Tukey's test as a post hoc analysis to evaluate the pairwise group differences. A *P*-value less than 0.05 was considered to be significant.

# Data Availability

This study does not include any data deposited in external repositories.

**Table 1.   Gene-specific primers designed for RT–PCR.**

| Gene name (gene symbol) | Forward primer | Reverse primer |
|---|---|---|
| V1a receptor (*AVPR1A*) | 5'-ctgtgtcagcagcgtgaagt-3' | 5'-gtttgttgggcttcgattgt-3' |
| V1b receptor (*AVPR1B*) | 5'-ccctgatgaagattccacca-3' | 5'-ctcccagctctcatttccag-3' |
| V2 receptor (*AVPR2*) | 5'-tgcctcctcctacatgatcc-3' | 5'-agggcaatccaggtgacata-3' |
| oxytocin receptor (*OXTR*) | 5'-ttcttcgtgcagatgtggag-3' | 5'-gccatcacctaggagcagag-3' |

# Supplementary Information

# Acknowledgements

We would like to thank Ayumi Izawa and Yuki Oyama for their technical assistance. This work was supported in part by research grants from the Scientific Fund of the Ministry of Education, Science, and Culture of Japan (24790259, 21K09477). The study was also supported by Kowa Life Science Foundation, the Japan Keirin Autorace Foundation, and the Promotion and Mutual Aid Corporation for Private Schools of Japan. We would like to thank Editage (www.editage.com) for English-language editing.

## Author Contributions

H Tsuchiya: conceptualization, funding acquisition, investigation, methodology, writing—original draft, and project administration.
M Fujinoki: investigation, methodology, and writing—original draft.
M Azuma: supervision, validation, and project administration.
T-a Koshimizu: funding acquisition, supervision, validation, and project administration.

## Conflict of Interest Statement

The authors declare that they have no conflict of interest.

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
