## [Reviewer comments · Life Science Alliance]

Life Science Alliance

Vasopressin V1a receptor and oxytocin receptor regulate murine sperm motility differently

Hiroyoshi Tsuchiya, Masakatsu Fujinoki, Morio Azuma and Taka-aki Koshimizu

DOI: <https://doi.org/10.26508/lsa.202201488>

Corresponding author(s): *Dr. Hiroyoshi Tsuchiya (Jichi Medical University)*

Review Timeline:

Submission Date:	2022-04-19
Editorial Decision:	2022-06-10
Revision Received:	2022-12-05
Editorial Decision:	2023-01-03
Revision Received:	2023-01-06
Accepted:	2023-01-06

Scientific Editor: Novella Guidi

Transaction Report:

June 10, 2022

Re: Life Science Alliance manuscript #LSA-2022-01488-T

Dr. Hiroyoshi Tsuchiya
Jichi Medical University
Pharmacology
3311-1 Yakushiji
Shimotsuke, Tochigi, 329-0498
Japan

Dear Dr. Tsuchiya,

Thank you for submitting your manuscript entitled "Mice with posterior pituitary hormone defect show aberrant sperm motility" to Life Science Alliance. The manuscript was assessed by expert reviewers, whose comments are appended to this letter. We invite you to submit a revised manuscript addressing the Reviewer comments.

Thank you for this interesting contribution to Life Science Alliance. We are looking forward to receiving your revised manuscript.

Sincerely,

B. MANUSCRIPT ORGANIZATION AND FORMATTING:

Reviewer #1 (Comments to the Authors (Required)):

Hiroyoshi Tsuchiya et al. (Hiroyoshi Tsuchiya, Masakatsu Fujinoki, Morio Azuma, and Taka-aki Koshimizu) present a study untitled: "Mice with posterior pituitary hormone defect show aberrant sperm motility". In this study, the authors first focused on the distribution and function of V1a, an AVP receptor. They showed a high expression of the V1a receptor in Leydig cells of the testes and narrow/clear cells in the epididymis by In-situ hybridization. In comparison, they showed that the pattern of expression of V1a was different with that of oxytocin receptor (OTR). They further analyzed the function of V1a using a deficient mouse model in this particular receptor and showed persistent motility and highly proportional hyperactivation in spermatozoa from V1a-deficient mice. They followed by testing the action of atosiban, and showed that the hyperactivation decreased the antagonist, atosiban-induced blockade of OTR. The authors suggest that V1a antagonists may represent therapeutic drugs for male infertility.

This is a good work on an important subject. As the authors mentioned the lack of selective antibodies for V1a is impeding research in this domain and the distinct mechanisms that these receptors may play. The work on a deficient mouse model was the answer. This is also interesting in that it highlighted the role of V1a in the male reproductive physiology. Although the overall study is quite interesting in the field I have few remarks and suggestions that may help the study.

- 1- In some part of the study, such as in the introduction, the authors cite very old references. I like that as it shows the original data. However, it would be appreciated to cite recent data confirming these data. For example: "the cells in male reproductive tissues that express the AVP receptor remains largely unknown (Maggi et al., 1987; Phillips et al, 1990). Have the years of experience since 1990 change that or not? Please, update the references if possible.
- 2- The generation of mice deficient for a particular gene is a 20-year-old technique and became a standard tool in mouse genetics which may help assign a function or few of them to each gene. However, in many cases some phenotypes are masked. It is not the case here, which is great, but did you find some unexpected unchanged phenotypes. If yes, please indicate them.
- 3- There have been few instances where the authors describe the data from hybridization as V1a receptor. It should be rectified to mRNA.
- 4- There has been multiple evidence for a species difference in the expression of OTR and vasopressin receptors, at least in parts of the brain. Could the authors discuss, although briefly this point?
- 5- There is also evidence from these animal studies that oxytocin and vasopressin and their receptors may play a role in human social relationships (opposite to each other). Did the authors observe any change in social interaction in their mouse model?

Reviewer #2 (Comments to the Authors (Required)):

- 1- Please explain how the number of animals was arrived at. Provide details of any sample size calculation used.
- 2- Please explain how the number of animals was arrived at. Provide details of any sample size calculation used.
- 3- Explain the inclusion and exclusion criteria in the methods.

Reviewer #3 (Comments to the Authors (Required)):

The authors address (mainly) the roles of the V1a receptor in male reproduction. In addition they touch on OTR in the epididymis.

This is an interesting, yet challenging topic, due to the similarities between AVP and OT and their respective receptors. The high degree of homology between the receptors explains also the poor selectivity of commercially available antibodies and the lack of information about actions of these hormones, especially in the male. In this study, the authors sought to overcome some of these obstacles by using a deficient mouse model and in-situ hybridization (ISH, for V1a and OTR).

The results, in essence, indicate a number of abnormalities, including mainly however the morphology of the epididymis and somehow surprisingly altered sperm motility in V1a-deficient mice. The authors claim that spermatozoa from the V1a-deficient mice may be more active than those from WT mice. Most likely this could be due to an alteration in the environment of the epididymis, but the actual reason(s) remain(s) unclear. This is a weakness of the paper.

The paper, in general, reads rather well (only few language related-problems), but there are a number of other points that remain unclear to me.

1. The mutant mice were described years ago. What else is known about the mice and what else has been described since the first report? Have the authors checked for impairments of fertility? Were such studies done at all? Was the hypothalamic pituitary gonadal axis thoroughly examined? Other than smaller size (body and testes), I can not find such information.
2. In the Results the authors write about the adrenal expression of V1a. I find the information provided there confusing, as in rodents the main glucocorticoid is corticosterone! This should be clarified. Also, I am not sure how relevant this information is in the context of this study.
3. I am afraid that I can not find information on animal numbers and on how often which specific experiment was repeated. Such information must be added.
4. Also, I do not find information on control experiments, especially controls for immunohistochemistry are missing. Such controls (e.g. IgG, non-immune serum) are crucial! In the same line, detailed information of the sequence of the probes used for ISH also are important!
5. Regarding the atosiban treatment, I am afraid that I can not fully follow the experiment: Mice were injected apparently daily for 14 d. Epididymis sperm were then analyzed for motility and hyperactivation. Such experiments must be well controlled. What was the control for this experiment? How were potential side-effects evaluated? What conclusion can be drawn from the experiments described given that atosiban is an apparently exclusive OTR antagonist in mice? Obviously, I miss some crucial piece of information, yet I understand from the data shown that OTR and V1a have distinct localizations in the epididymis. It would be desirable to investigate the consequences of actions of receptor activation further. Are pH changes involved?
6. Language issues:
Examples:that does not cross the OTR could lead tofurther studies in humans is necessary because male reproductive tissues
Also, the title should be reconsidered and focused, as a receptor, rather than the hormone was knocked out.

In summary, this is - in general- an interesting topic and (possibly) reasonable approaches to tackle this topic are presented. Yet, without seeing the mentioned controls and without additional information, I can not evaluate the quality of the results and hence of the paper, in general.

Reviewer #1 (Comments to the Authors (Required)):

Hiroyoshi Tsuchiya et al. (Hiroyoshi Tsuchiya, Masakatsu Fujinoki, Morio Azuma, and Taka-aki Koshimizu) present a study untitled: "Mice with posterior pituitary hormone defect show aberrant sperm motility".

In this study, the authors first focused on the distribution and function of V1a, an AVP receptor. They showed a high expression of the V1a receptor in Leydig cells of the testes and narrow/clear cells in the epididymis by In-situ hybridization. In comparison, they showed that the pattern of expression of V1a was different with that of oxytocin receptor (OTR). They further analyzed the function of V1a using a deficient mouse model in this particular receptor and showed persistent motility and highly proportional hyperactivation in spermatozoa from V1a-deficient mice. They followed by testing the action of atosiban, and showed that the hyperactivation decreased the antagonist, atosiban-induced blockade of OTR. The authors suggest that V1a antagonists may represent therapeutic drugs for male infertility.

This is a good work on an important subject. As the authors mentioned the lack of selective antibodies for V1a is impeding research in this domain and the distinct mechanisms that these receptors may play. The work on a deficient mouse model was the answer. This is also interesting in that it highlighted the role of V1a in the male reproductive physiology.

Although the overall study is quite interesting in the field I have few remarks and suggestions that may help the study.

1- In some part of the study, such as in the introduction, the authors cite very old references. I like that as it shows the original data. However, it would be appreciated to cite recent data confirming these dat. For example: "the cells in male reproductive tissues that express the AVP receptor remains largely unknown (Maggi et al., 1987; Phillips et al, 1990). Have the years of experience since 1990 change that or not? Please, update the references if possible.

We have added references carried out after 1990; Gupta *et al*, 2008 and Mewe *et al*, 2007. These articles are indirect experiments using specific binding substances instead of direct tissue distribution experiments, such as *in situ hybridization* analysis. To the best of our knowledge, no other experiments have examined the vasopressin receptor distributions in the male reproductive organs, since little attention has been paid recently to the role of vasopressin and its receptors in the male reproductive organs. That is why our research is meaningful.

2- The generation of mice deficient for a particular gene is a 20-year-old technique and became a standard tool in mouse genetics which may help assign a function or few of them to each gene.

However, in many cases some phenotypes are masked. It is not the case here, which is great, but did you find some unexpected unchanged phenotypes. If yes, please indicate them.

We have not made any observations beyond those presented in this paper; therefore, we do not have any specific results to show. In a previous study, we reported no significant changes in litter size, even if the genetically V1a-deficient males are mated to wild type female (Tsuchiya *et al*, 2020). This result means that improved sperm motility does not increase the number of births beyond the upper limit.

3- There have been few instances where the authors describe the data from hybridization as V1a receptor. It should be rectified to mRNA.

We have rectified the descriptions.

4- There has been multiple evidence for a species difference in the expression of OTR and vasopressin receptors, at least in parts of the brain. Could the authors discuss, although briefly this point?

We have added information on the species difference with regard to the brain and the epididymis, in the discussion.

5- There is also evidence from these animal studies that oxytocin and vasopressin and their receptors may play a role in human social relationships (opposite to each other). Did the authors observe any change in social interaction in their mouse model?

We understand the fact that vasopressin and oxytocin are closely related to social behavior; however, because our studies were not analyzed in a system that can detect any social behavior, we cannot clarify whether there are further abnormalities in the V1a receptor-deficient mice. In an overview of the sexual behavior, the deficient mice are fertile and the latency of the ejaculation was similar to that of the WT mice. However, it has been observed from preliminary research that the probability of a failed pregnancy in successfully mated female mice was 28% in V1a-deficient mice compared to 8% in WT mice. Although interesting, this result has not been included in this manuscript because it would be a marked departure from the theme of the current paper.

Reviewer #2 (Comments to the Authors (Required)):

1- Please explain how the number of animals was arrived at. Provide details of any sample size calculation used.

We have responded to this comment based on Figure 5. The number of animals was adjusted to be as equal as possible within the comparison group, using males born from three or more mothers. We

have added the information in the Materials and Methods section. In addition, the number of samples has been added to the figure legend.

2- Please explain how the number of animals was arrived at. Provide details of any sample size calculation used.

We have responded based on Figure 6. The number of animals was adjusted to be as equal as possible within the comparison group, using males born from three or more mothers. In addition, care was taken to ensure that littermates were assigned to both the control and Atosiban-treatment groups. We have added the information in the Materials and Methods section. In addition, the number of samples has been added to the figure legend.

3- Explain the inclusion and exclusion criteria in the methods.

We have responded to this comment with regard to counting of sperm. We defined motile sperm as those with flagellar movement and excluded all others from our calculations. We have added the information in the Materials and Methods section.

Reviewer #3 (Comments to the Authors (Required)):

The authors address (mainly) the roles of the V1a receptor in male reproduction. In addition they touch on OTR in the epididymis.

This is an interesting, yet challenging topic, due to the similarities between AVP and OT and their respective receptors. The high degree of homology between the receptors explains also the poor selectivity of commercially available antibodies and the lack of information about actions of these hormones, especially in the male. In this study, the authors sought to overcome some of these obstacles by using a deficient mouse model and in-situ hybridization (ISH, for V1a and OTR).

The results, in essence, indicate a number of abnormalities, including mainly however the morphology of the epididymis and somehow surprisingly altered sperm motility in V1a-deficient mice. The author claim that spermatozoa from the V1a-deficient mice may be more active than those from WT mice. Most likely this could be due to an alteration in the environment of the the epididymis, but the actual reason(s) remain(s) unclear. This is a weakness of the paper.

The paper, in general, reads rather well (only few language related-problems), but there are a number of other points that remain unclear to me.

1. The mutant mice were described years ago. What else is known about the mice and what else has been described since he first report? Have the authors checked for impairments of fertility? Were such studies done at all? Was the hypothalamic pituitary gonadal axis thoroughly examined? Other than smaller size (body and testes), I can not find such information.

We have added the information about V1a-deficient mice in Introduction. In brief, various abnormalities have been identified in V1a-deficient mice, including cardiovascular system abnormalities such as decreased heart rate and hypotension, impaired social interaction, impaired glucose homeostasis, insufficient water metabolism, circadian rhythm misalignment, and impaired female reproductive physiology. In a previous study, we observed decreased litter size when V1a-deficient mice were the mother. When we used the V1a-deficient mice as the father, we could not detect significant abnormality in litter size. We speculate that redundancy rescued the abnormality of paternal origin.

We also measured the weights of seminal vesicle, preputial gland, and prostate gland. When the weights were adjusted according to individual body weights, there was no difference in preputial gland weight, in addition to testes and epididymis weight. However, the seminal vesicle in V1a receptor-deficient mice was significantly larger than that in the WT, and the prostate gland in V1a-deficient mice was significantly smaller. The results have been presented in Figure S1 A-C. We also measured serum testosterone levels. No significant differences in serum testosterone levels were observed between V1a-deficient mice and WT mice in our experimental setting.

2. In the Results the authors write about the adrenal expression of V1a. I find the information provided there confusing, as in rodents the main glucocorticoid is corticosterone! This should be clarified. Also, I am not sure how relevant this information is in the context of this study.

Of course, in rodents, the main glucocorticoid is corticosterone. However, in addition to the production of glucocorticoids and mineralocorticoids, adrenal glands produce adrenergic androgens such as dehydroepiandrosterone (DHEA) and dehydroepiandrosterone sulfate (DHEAS), which can work as ligands to the androgen receptor in primates. In rodents, adrenergic androgen secretions are absent. This species difference has precluded the detailed analysis of the roles of adrenergic androgens. In this paper, we have shown, for the first time, to the best of our knowledge, the distribution of the *Avpr1a* in the murine adrenal gland, and revealed that the *Avpr1a* signals were scattered throughout the adrenal gland. In the future, we would like to compare the present results with expression in adrenal androgen-producing species and clarify the differences in distribution of V1a receptors between the sexes.

For clarity, some text from the Results has been moved to the Discussion, and further information has been added.

3. I am afraid that I can not find information on animal numbers and on how often which specific experiment was repeated. Such information must be added.

Thank you for your comment. The number of animals and repeated experiments were missing in the qualitative experiments, such as the β -galactosidase staining, *in-situ* hybridization, and immunohistochemistry. We have added the information in the Materials and Methods.

We have also added the number of data of the quantitative experiments, such as in Fig 5 and 6, in the figure legends.

4. Also, I do not find information on control experiments, especially controls for immunohistochemistry are missing. Such controls (e.g. IgG, non-immune serum) are crucial! In the same line, detailed information of the sequence of the probes used for ISH also are important!

We have added the negative control of IHC, as Figure S3. We were unable to represent the detailed information of the sequence of the probes because the provider company, Advanced Cell Diagnostics, has made it proprietary information, and disclosure is not permitted. However, researchers can reproduce the experiments, if they use the same probes; no. 418061 for *Avpr1a* and no. 402651-C2 for *Oxtr*.

5. Regarding the atosiban treatment, I am afraid that I can not fully follow the experiment: Mice were injected apparently daily for 14 d. Epididymis sperm were then analyzed for motility and hyperactivation. Such experiments must be well controlled. What was the control for this experiment? How were potential side-effects evaluated? What conclusion can be drawn from the experiments described given that atosiban is an apparently exclusive OTR antagonist in mice? Obviously, I miss some crucial piece of information, yet I understand from the data shown that OTR and V1a have distinct localizations in the epididymis. It would be desirable to investigate the consequences of actions of receptor activation further. Are pH changes involved?

In this experiment, the vehicle (saline) group was used as the control. Atosiban is a safe clinical drug with few serious adverse effects when used as a therapeutic agent in humans (The most common side effect of atosiban is nausea [10%]). Please refer to the European Medicines Agency website (<https://www.ema.europa.eu/en/medicines/human/EPAR/tractocile>) and other sources. No apparent abnormalities were observed in our experiment. Body weights were measured before and after the experiment and no significant differences were found between the two groups (pre-treatment: saline 30.9 ± 0.29 g, atosiban 30.88 ± 0.78 g, post-treatment: saline 31.34 ± 0.28 g, atosiban 31.56 ± 0.73 g).

Since atosiban inhibited sperm motility in mice, oxytocin is expected to activate motility via its specific receptor. However, it is unclear which cells are important, as the expression distribution

analysis showed that unlike V1a, oxytocin receptors were expressed in a wide variety of cell types of the epididymis. Further analysis is required.

Additional experiments were conducted to analyze the relationship between V1a receptor and pH. Unfortunately, cells expressing V1a receptor in the epididymis are scattered, making it difficult to use them to investigate changes in pH. Therefore, the pH measurement experiment was performed in the cell line of the kidney, which is close in origin to the epithelium of the epididymis (Hinton BT and Turner TT. 1988 *News Physiol Sci*). The use of cell lines with uniform expression characteristics has the advantage of making it easier to detect changes in pH during agonist stimulation. In this report, Caki-2 cells, which have been used in previous reports, were used for the pH measurement experiment (Al-bataineh et. al. 2016 *Am J Physiol Renal Physiol*). The experimental method was also based on that in previous reports.

Experimental results showed that AVP stimulation increased pH. This effect was abolished by V1a antagonist RO5028442, suggesting that it is mediated by the V1a receptor. In other words, V1a receptor activation is thought to suppress proton release. The activation of sperm motility observed in V1aKO is expected to result from the invalidation of proton release inhibition induced by AVP within the epididymis, causing acidification of the duct, which in turn affects sperm motility function. Future studies are required to determine the actual pH changes in the epididymal tubes and the associated changes in genes and proteins.

6. Language issues:

Examples:that does not cross the OTR could lead tofurther studies in humans is necessary because male reproductive tissues

Also, the title should be reconsidered an focused, as a receptor, rather than the hormone was knocked out.

Thank you. We have corrected the mistakes.

We have changed the title to “Vasopressin V1a receptor and oxytocin receptor regulate murine sperm motility differently.” to make it clear that we are focusing on receptors.

In summary, this is - in general- an interesting topic and (possibly) reasonable approaches to tackle this topic are presented. Yet, without seeing the mentioned controls and without additional information, I can not evaluate the quality of the results and hence of the paper, in general.

January 3, 2023

RE: Life Science Alliance Manuscript #LSA-2022-01488-TR

Dr. Hiroyoshi Tsuchiya
Jichi Medical University
Pharmacology
3311-1 Yakushiji
Shimotsuke, Tochigi, 329-0498
Japan

Dear Dr. Tsuchiya,

Thank you for submitting your revised manuscript entitled "Vasopressin V1a receptor and oxytocin receptor regulate murine sperm motility differently". We would be happy to publish your paper in Life Science Alliance pending final revisions necessary to meet our formatting guidelines.

- please address Reviewer 3's remaining comments
- please upload your main figures as single files; these will be displayed in-line in the HTML version of your paper, so please provide them as single page files (Figures 2 and 5 currently span 2 pages); we do not have a limit on the number of main figures and these can be split if necessary for space
- please add ORCID ID for corresponding author-you should have received instructions on how to do so
- please add the Twitter handle of your host institute/organization as well as your own or/and one of the authors in our system

A. FINAL FILES:

B. MANUSCRIPT ORGANIZATION AND FORMATTING:

Sincerely,

Reviewer #3 (Comments to the Authors (Required)):

I have re-reviewed the paper. My points of concern, raised after reading the original version, were in general adequately addressed and the paper has improved.

It contains important new information. However, a few points still need attention:

- The following sentence in the abstract should be changed ..." In contrast, hyperactivation decreased antagonist atosiban-mediated blocking of the OTR. ..."
- The authors mention "controls" for IHC in their response letter and also in the paper refer to S3. ("...We have added the negative control of IHC, as Figure S3...") However, they do not mention the nature of the control! This should be specified.

Reviewer #3 (Comments to the Authors (Required)):

I have re-reviewed the paper. My points of concern, raised after reading the original version, were in general adequately addressed and the paper has improved.

It contains important new information. However, a few points still need attention:

1- The following sentence in the abstract should be changed ..." In contrast, hyperactivation decreased antagonist atosiban-mediated blocking of the OTR. ..."

We have improved the description for clarity.

2- The authors mention "controls" for IHC in their response letter and also in the paper refer to S3.("...We have added the negative control of IHC, as Figure S3...") However, they do not mention the nature of the control! This should be specified.

We missed to describe the details. We have added the information about the negative control in Materials and Methods and the legend of Fig. S3.

January 6, 2023

RE: Life Science Alliance Manuscript #LSA-2022-01488-TRR

Dr. Hiroyoshi Tsuchiya
Jichi Medical University
Pharmacology
3311-1 Yakushiji
Shimotsuke, Tochigi, 329-0498
Japan

Dear Dr. Tsuchiya,

Thank you for submitting your Research Article entitled "Vasopressin V1a receptor and oxytocin receptor regulate murine sperm motility differently". It is a pleasure to let you know that your manuscript is now accepted for publication in Life Science Alliance. Congratulations on this interesting work.

DISTRIBUTION OF MATERIALS:

Again, congratulations on a very nice paper. I hope you found the review process to be constructive and are pleased with how the manuscript was handled editorially. We look forward to future exciting submissions from your lab.

Sincerely,
